# Elevated IL-6 and IL-22 in Early Pregnancy Are Associated with Worse Disease Course in Women with Inflammatory Bowel Disease

**DOI:** 10.3390/ijms231810281

**Published:** 2022-09-07

**Authors:** Richard Y. Wu, Karren Xiao, Naomi Hotte, Parul Tandon, Yesmine Elloumi, Lindsy Ambrosio, Garett Dunsmore, Shokrollah Elahi, Karen I. Kroeker, Levinus A. Dieleman, Karen L. Madsen, Vivian Huang

**Affiliations:** 1Department of Medicine, University of Toronto, Toronto, ON M5T 2S8, Canada; 2Division of Gastroenterology and Hepatology, Mount Sinai Hospital, Toronto, ON M5T 2S8, Canada; 3Division of Gastroenterology, University of Alberta, Edmonton, AB T6G 2R3, Canada; 4Department of Medical Microbiology and Immunology, Faculty of Medicine and Dentistry, University of Alberta, Edmonton, AB T6G 2R3, Canada; 5Department of Dentistry, Faculty of Medicine and Dentistry, University of Alberta, Edmonton, AB T6G 2R3, Canada; 6Department of Oncology, University of Alberta, Edmonton, AB T6G 2R3, Canada; 7Li Ka Shing Institute of Virology, University of Alberta, Edmonton, AB T6G 2R3, Canada

**Keywords:** UC, CD, pregnancy, cytokines, IL-6, IL-22

## Abstract

Inflammatory bowel diseases (IBD), including Ulcerative Colitis (UC) and Crohn’s disease (CD), are inflammatory conditions of the intestinal tract that affect women in their reproductive years. Pregnancy affects Th1- and Th2-cytokines, but how these changes occur during pregnancy in IBD is unclear. We performed a longitudinal profiling of serum cytokines in a cohort of 11 healthy pregnant women and 76 pregnant women with IBD from the first trimester of pregnancy to the first 12 months post-partum. Participants were monitored for biochemical disease activity (C-reactive protein [CRP] and fecal calprotectin [FCP]) and clinical activities. Maternal cytokines were measured using ELISA. We identified changes in Th1 and Th17 cytokines throughout pregnancy in healthy pregnant women. During pregnancy, maternal serum cytokine expressions were influenced by IBD, disease activity, and medications. Active UC was associated with an elevation in IL-21, whereas active CD was associated with elevated IFN-γ, IL-6, and IL-21. Interestingly, T1 serum cytokine levels of IL-22 (>0.624 pg/mL) and IL-6 (>0.648 pg/mL) were associated with worse IBD disease activity throughout pregnancy in women with UC and CD, respectively. This shows serum cytokines in pregnancy differ by IBD, disease activity, and medications. We show for the first time that T1 IL-22 and IL-6 correlate with IBD disease course throughout pregnancy.

## 1. Introduction

Inflammatory bowel disease (IBD) is a group of chronic inflammatory conditions affecting the gastrointestinal tract and includes two primary entities: ulcerative colitis (UC) and Crohn’s disease (CD) [1]. The prevalence of IBD is highest in early adulthood between the second and fourth decade of life, thereby affecting many women in their reproductive years [2]. Pregnancy generally produces an immunotolerant state [3] and has been shown to reduce long-term relapses [4]. However, while patients with stable disease at conception have comparable relapse rates to non-pregnant patients [5], those with active disease are much more likely to persist throughout pregnancy [6,7] and carry a higher risk of preterm delivery, low-birth-weight infants, and adverse obstetrical outcomes [8]. Because of these concerns, it is recommended for women with IBD contemplating pregnancy to achieve remission before conception [2]. However, the fundamental understanding of why IBD flares up in some women during pregnancy is largely unknown.

The immunopathogenesis of IBD is driven by a complex cascade of innate and adaptive immune responses in the intestinal mucosa [9]. CD is traditionally thought to be mediated by Th1- (TNF-α, IFN-γ, IL-1β) and Th17-associated (IL-17, IL-22) cytokines [10], whereas UC is thought to be primarily driven by a Th2-response (IL-4, IL-5, and IL-13) [11]. However, the immunological landscape in healthy women fluctuates throughout pregnancy and follows a bimodal pattern whereby there is an initial Th1-response at implantation that shifts towards Th2 throughout pregnancy until a rebound in Th1-response at parturition [12]. These dynamic cytokine changes facilitate a tightly regulated immunotolerant state at the maternal–fetal interface to optimize implantation, growth, and eventual partition [13]. However, to what extent IBD impacts the pregnancy-related immune response, and vice versa, is currently unclear. Previously, we demonstrated that cord blood samples from pregnant women with IBD had a significantly lower abundance of tolerogenic CD71^+^ erythroid cells compared to healthy pregnant women, and this also correlated with a difference in functional phenotype, including lower TGF-β and elevated IL-6 production [14]. We have also reported that a lower proportion of CD71^+^ erythroid cells was associated with worse pregnancy outcomes [15], suggesting that the maternal immune changes in IBD may significantly influence pregnancy-related outcomes in IBD. However, to date, few studies have examined if (a) IBD influences pregnancy-related cytokine changes and (b) whether a difference in cytokine patterns may influence the course of IBD disease trajectory or clinical outcomes at delivery.

To characterize the cytokine changes in pregnant women with IBD, herein we performed a pilot study to measure the maternal serum cytokines longitudinally from trimester 1 until 12 months post-partum in a cohort of 76 pregnant women with IBD. To interrogate the factors influencing host cytokine changes, we stratified cytokine changes based on IBD status, IBD subtype, disease flare-up, and medications used at the time of sampling. Furthermore, to explore the clinical utility of cytokine levels, we correlated serum cytokines with disease activity across time points and tested the predictive utility of early pregnancy serum cytokine changes against future flare-ups during pregnancy as well as pregnancy-related outcomes.

## 2. Results

### 2.1. Baseline Demographics and Clinical Characteristics

Demographic details of the study cohort can be found in Table 1. A total of 87 women were included in the current study: 11 healthy women, 29 women with CD, and 47 women with UC. The median age of healthy women was 31, and 31 and 30 for patients with CD and UC, respectively (Table 1). The groups were similar in the age of IBD diagnosis, ethnicity, employment status, education status, smoking, alcohol use, and prior pregnancy histories. Over the course of pregnancy, there was a difference observed in clinical disease status (measured using partial Mayo for UC or mHBI for CD), whereby more women with CD were noted to be in a clinical flare (Table 1). Overall, baseline demographics were similar across both groups. The complete details of the pregnancy-related outcomes are found in Table 2, and medications are found in Table 3. Between patients with CD versus UC, there were higher rates of pre-eclampsia (*p* = 0.02) amongst the pregnant women with CD versus UC. Notably, the rates of preterm births, placental abnormalities, and delivery outcomes were comparable across both groups. There was a trend for lower rates of vaginal deliveries but higher planned C-sections amongst women with CD, although this did not reach statistical significance.

### 2.2. Pregnancy-Related Changes in Th1, Th2, and Th17 Cytokines

To determine the longitudinal effects of pregnancy on host serum cytokines, we collected serum samples from 11 healthy women across 6 time points: T1, T2, T3, PP3, PP6, and PP12. Overall, there was an increase in select Th1 cytokines throughout pregnancy with a sustained increase in TNF-α and IL-1β that peaked at PP3 (Figure 1A). IFN-γ levels were elevated at T2 and remained sustained throughout pregnancy (Figure 1A). On the other hand, Th2 cytokines were relatively stable, but at PP3-12, there was an increase in IL-4, and IL-10, whereas IL-5 and IL-13 levels remained unchanged (Figure 1B). Among Th17 cytokines, IL-21, IL-22, and IL-23 showed sustained increase throughout pregnancy and peaked at T3, whereas IL-17 peaked at PP3 (Figure 1C). Among inflammatory cytokines, CRP and IL-27 were consistently elevated across pregnancy and tapered post-partum amongst healthy women. Chemokines IL-6 and IL-8 expression, on the other hand, remained stable during pregnancy, but IL-8 peaked at PP3 (Figure 1D).

### 2.3. Serum Cytokine Changes by IBD during Pregnancy

To compare the effects of IBD on serum cytokines, we next analyzed the serum cytokines from women with IBD throughout pregnancy. Overall, 31 patients provided serum samples at T1, 51 provided samples at T2, 60 at T3, 54 at PP3, 34 at PP6, and 22 at PP12. The IBD characteristics, including active medication use and biochemical disease flare-up at the time of serum sampling, can be found in Figure 2A. To compare the overall differences in cytokine changes during and after pregnancy, we performed principal component analysis (PCA) at T2 and PP3, given that the highest number of serum samples were available at these time points. This showed overall poor separation between healthy control women versus patients with either CD or UC both during (Figure 2B) and after pregnancy (Figure 2C). To further distinguish the time-specific differences in cytokine changes by IBD subtype, we analyzed all cytokines based on fold changes compared to healthy women and tested time-specific differences throughout pregnancy. We noted that while there were significant alterations in specific cytokines, most cytokines we measured remained unchanged during pregnancy. CD was associated with an elevation of IL-2 at T2, an increase in IL-27, and a decrease in IL-21 at T3 (Figure 2D). On the other hand, compared to healthy women, women with UC had higher IL-23 and lower IL-6 at T1 and decreased IL-27 at T3 (Figure 2D). However, after delivery, both women with UC and CD demonstrated completely different cytokine profiles compared to those seen during pregnancy. To understand the dynamic changes in cytokine expression, the cytokines with the largest variation by IBD status were IL-6, IL-22, and IL-21, and interestingly, CRP was among the cytokines with the lowest variation by IBD status. Pregnant women with CD had higher IL-6 levels than both healthy women and those with UC (Figure 2E). In contrast, IL-22 levels were higher in both pregnant women with UC or CD but were higher post-partum in women with UC (Figure 2F). IL-21 was higher in pregnant women with CD in the early trimesters but was more elevated in women with UC during the post-partum period (Figure 2G). In contrast, no large changes in CRP expression were noted during and after pregnancy (Figure 2H).

### 2.4. Difference in Serum Cytokines between Pre-Partum and Post-Partum Periods

To further interrogate the changes in cytokines across time points, next, we explored the impact of delivery on cytokine changes in women with CD versus UC. Among women with CD, we noted that, with delivery, there was a significant decrease in IL-27 (Figure 3A), IL-21 (Figure 3B), IL-23 (Figure 3C), and an increase in IL-12p40 (Figure 3D). Similarly, in patients with UC, we noted post-partum there was also a decrease in IL-27 (Figure 3E), IL-21 (Figure 3F), and an increase in IL-12p40 (Figure 3H). In addition, IFN-γ (Figure 3G) and IL-17 (Figure 3I) were both elevated specifically for women with UC.

### 2.5. Serum Cytokine Expressions Stratified by Medications and Disease Activity during Pregnancy

In order to assess the impact of medications and disease activity, we re-analyzed the serum cytokines from all time points during pregnancy based on the medications used and the disease activity at the time of collection. In pregnant women with CD, patients on either non-biologics (5-ASA, azathioprine) or biologics had lower levels of CRP and TNF-α, whereas IL-2 was only decreased by non-biologics (Figure 4A). On the other hand, in women with UC, patients on biologics showed higher levels of CRP and IFN-γ (Figure 4B). To also investigate the effect of active disease on cytokine expressions, we stratified patients based on active biochemical disease and IBD subtype (FCP ≥ 250 µg/g or CRP ≥ 10 mg/L). Interestingly, a total of 12 women with CD had flare-ups during pregnancy, and this was associated with an increase in IFN-γ, IL-21, and IL-6 (Figure 4C). On the other hand, a total of 19 women with UC flared up during pregnancy, and this was associated with an increase in IL-21 at the time of flare-up (Figure 4D). To confirm these results, we also performed correlation coefficients between biochemical disease activity and serum cytokines at the time of flare-up (Figure 4E). This showed that active flare-up was correlated with changes in TNF-α, IL-5, CRP, and IL-6 in patients with CD, whereas flare-up was correlated with TNF-α, IL-21 and CRP in patients with UC. Interestingly, during post-partum periods, active disease flare-up was not correlated with any changes in serum cytokines for women with CD or UC, respectively (Appendix A).

### 2.6. Impact of T1 Cytokines on Course of IBD and Pregnancy-Related Outcomes

To understand the clinical importance of the detected cytokines on the course of IBD, we performed ROC curves using cytokines previously found to be most influenced by disease status. Specifically, T1 cytokines were tested against T2 flare-up rates and T2 cytokines against T3 flare-up rates for patients with UC and CD, respectively. Across all tested time points, the only time point that generated statistically significant ROC models was at T1 (only the two significant models are shown). For women with UC, the T1 serum level of IL-22 predicted T2 disease activity (*p* = 0.02, Figure 5A) using a cut-off threshold of 0.624 pg/mL (Figure 5B). To test the impact of high versus low IL-22 on the clinical course of IBD using this cut-off, we analyzed the remission curves of women with UC separated based on T1 IL-22 (Figure 5C), which showed that women with T1 IL-22 > 0.624 pg/mL were more likely to flare-up during pregnancy (*p* < 0.0001, N = 12). On the other hand, for women with CD, the T1 serum level of IL-6 was the cytokine predictive of T2 disease activity (*p =* 0.04, Figure 5D) using a cut-off threshold of 0.648 pg/mL (Figure 5E). Similar to patients with UC, clustering patients based on T1 serum IL-6 levels completely predicted the course of IBD during pregnancy (*p* < 0.0001, N = 15, Figure 5F). Furthermore, to interrogate the associations between T1 cytokines and pregnancy-related outcomes, we correlated the T1 cytokine expressions with pregnancy and delivery outcomes (Figure 5G and Appendix A). In women with UC, T1 serum TNF-α, IFN-γ, IL-12p70, IL-21, and IL-17 expression significantly correlated with the mode of delivery. On the other hand, for women with CD, T1 serum IL-8 levels were correlated with the mode of delivery. We did not find any correlation between T1 cytokines to infant birthweight and length in patients with either UC or CD.

## 3. Discussion

Maintaining remission early in pregnancy is the current recommendation for the management of IBD in pregnancy [2]. However, specific host inflammatory cytokines naturally vary throughout the course of pregnancy, but whether this impacts women with IBD is largely unclear. To address this gap, our pilot study characterized the longitudinal changes in serum cytokines from T1 to post-partum month 12. Using a combination of longitudinal comparisons and multivariate modeling, we demonstrate that (1) there are inherent fluctuations in CRP, and Th1/Th17 cytokines during a normal pregnancy; (2) serum cytokines are most impacted by IBD, IBD subtypes, delivery, as well as the medications used; (3) specific cytokine changes are correlated with disease activity in women with UC and CD; (4) T1 serum cytokines are correlated with the method of delivery and predict the course of disease in women with UC and CD, respectively. To our knowledge, this is the first study to date to link T1 serum cytokines with the course of IBD disease activity throughout pregnancy.

The use of serum cytokines in IBD has been well-studied for their ability to discriminate tissue inflammation during active disease [16,17,18]. Specifically, CD has previously been shown to be driven by a Th1/Th17 phenotype, whereas UC is predominantly a Th2/Th17-driven disease [1]. However, only a few studies have examined how pregnancy-associated cytokine changes alter this cytokine pattern and vice-versa. Recently, van der Giessen et al. measured serum cytokines of 46 women with IBD (31 with CD and 15 with UC) and 179 healthy women [19]. Similar to their study, we demonstrate that in healthy women, there is a sustained increase in IL-6 levels throughout pregnancy, which peaked at T3. Furthermore, for IBD-specific changes during pregnancy, our results also showed that there were overall fewer cytokine changes during pregnancy compared to post-partum, which supports the theory that conception induces an overall immunotolerant state in patients with IBD as postulated previously [13,19]. However, we further add to this growing body of work by demonstrating that while there are few changes in serum cytokines during pregnancy, significant variations occur when patients are stratified by medications used, active disease versus non-active disease, as well as disease subtype. Specifically, disease activity in CD is associated with an increase in IFN-γ, IL-21, and IL-6, whereas disease activity in UC is only associated with an increase in IL-21. This fits with previous reports whereby IL-6 and IFN-γ elevation have been correlated with CD disease severity [19,20,21]. Similarly, IL-21, which is a cytokine produced by differentiated CD4^+^T-cells, has been central to the mediation of Th1-response in CD [22] and is correlated with active disease in CD [23]. IL-21 facilitates colitogenic phenotypes in UC, and IL-21^−/−^ knockout mice are protected against chemical-induced colitis [24]. Clinically, patients with UC have also previously been shown to have elevated IL-21 in rectal biopsies [25]. Therefore, our observations of persistent IL-21 elevations in both women with UC and CD during active flare-up fit with the existing relationship between IL-21 and the severity of intestinal inflammation. Additionally, the synchronous IL-6/IL-21 in our CD cohort also supports the ongoing hypothesis that IL-6 triggers the activation of IL-21 during CD pathogenesis [23].

To our surprise, the majority of serum cytokines remained stable throughout pregnancy. We could not identify any major shifts in Th1, Th2, or Th17-associated cytokine responses with IBD subtype or active disease. Instead, differences were only seen in a few select cytokine targets at key time points. This could be explained by the existing model whereby pregnancy induces an overall immunotolerant state, but may also suggest the emerging theory whereby cytokine networks are neither tissue- nor disease-specific and are dynamic over time depending on the stage of pathogenesis of the inflammatory disease [9]. However, further validations using much larger cohorts and cytokine targets are warranted to validate our results.

It is noteworthy that the serum cytokine profiles not only correlated with disease activity at the time of sampling, but early T1 cytokines also predicted IBD course throughout pregnancy. In fact, the separation of pregnant women based on T1 IL-22 (≥0.624 pg/mL) and IL-6 (≥0.648 pg/mL) was predictive of the disease course for UC and CD, respectively. IL-22 is an IL-10-family cytokine expressed by innate lymphoid cells and differentiated CD4^+^T cells [26]. Functionally, IL-22 is associated with intestinal epithelial barrier integrity and innate immune functions, including mucin expression, epithelial cell differentiation, and antimicrobial peptide expression [27]. Using in vivo models, IL-22 has been shown to be protective against pathogen-induced colitis, including *C. rodentium* [28] and *Salmonella typhimurium* [29], and facilitates the clearance of pathogenic fungi [30]. In parallel, IL-22 also facilitates intestinal regeneration and differentiation through direct actions on Lgr5+ stem cells within intestinal crypts. Our finding of elevated IL-22 in women with a worse course of IBD disease in pregnancy is congruent with the existing literature. In fact, IL-22 has previously been shown to be more elevated during active inflammation in IBD [31] and has been recently investigated for novel biologic therapy MEDI2070 [32,33]. Similarly, IL-6 is a pro-inflammatory cytokine associated with innate immune mucosal response, and its elevation is associated with severe complications in CD [34]. Recently, PF-04236921, an IL-6 neutralizing antibody, has also been investigated in patients with severe CD within the ANDANTE I and II trials [35]. In parallel, both IL-6 and IL-22 are associated with adverse pregnancy outcomes. For instance, elevation in IL-6 has been associated with pre-eclampsia, gestational diabetes, and hypertension [36]. Similarly, elevation in early gestation serum IL-22 is associated with recurrent pregnancy loss and spontaneous abortions [37]. The observed association between T1 IL-6 elevation and adverse IBD course in pregnancy fits with the existing literature. While it is tempting to speculate whether targeting IL-22 and IL-6 early T1 could improve pregnancy outcomes, much larger studies using larger sample sizes and diverse disease severities are warranted to confirm these preliminary results.

To our surprise, amongst the inflammatory mediators identified, one of the biomarkers least affected by IBD status during pregnancy was CRP. CRP is an acute inflammatory protein elevated in response to innate immune cytokine release and clinically has been validated as a routine clinical marker of inflammation in non-pregnant patients [38]. In our cohort, CRP was persistently elevated across time points in healthy women with no differences compared to either IBD subtype. This fits with previous reports showing that CRP levels increase naturally in healthy pregnant subjects compared to non-pregnant women [39,40]. Persistent elevations in CRP have also been correlated with preterm delivery [41] and worse maternal pregnancy-related outcomes [36,42]. While the precise mechanisms are unclear, it is currently postulated that this may be mediated by IL-6 [41,43] during the acute phase of acute inflammation. Our observation of also persistent elevations in IL-6 by late gestation also fits with this hypothesis. More importantly, our observations that CRP was not altered by either CD or UC question the clinical utility of CRP as a non-invasive method during pregnancy [44]. Instead, more targeted monitoring in precise time points in unique IBD subtypes may be warranted.

Although our study demonstrated interesting cytokine alterations by IBD, disease activity, and medication use, there are multiple limitations to our study. Foremost, while our cohort size of 76 women with IBD is the largest cohort size to date to serially monitor pregnant women with IBD, it lacks substantial heterogeneity in the patient population. As a pilot single-center study, the geographical catchment was limited, whereby the majority of patients were Caucasians. Secondly, while we attempted to serially monitor the cohort from T1 to PP12, there were significant losses-to-follow-up, particularly at later time points, which impeded cross-timepoint comparisons. As a result, select comparisons, such as cytokine comparisons between pregnancy and post-partum time points, were done without controlling for medications used, which makes it difficult to interpret whether the observed changes are truly due to pregnancy timing itself. Thirdly, the study timeline preceded the use of Vedolizumab and Ustekinumab, and therefore, the majority of our patients were on anti-TNF-α therapy. In addition, while we recorded the use of medications, drug levels were not measured for biologics. Given the lack of accessibility to drug level monitoring in Canada from 2014 to 2017, the patient drug levels could not be recorded in our study, which could inevitably affect the interpretation of the cytokine datasets. Furthermore, although maternal cytokines were assayed at the same time, we did not control for the timing of when maternal samples were drawn. This can cause variations, given that variations in serum cytokine can exist. In light of these limitations, although our pilot study is hypothesis-generating, future studies incorporating a larger and more diverse patient population with timepoint-specific drug monitoring are warranted to further substantiate our findings.

## 4. Materials and Methods

### 4.1. Study Design and Study Cohort

We performed a single-center prospective cohort study based at the University of Alberta (Edmonton, AB) from 2014 and 2017, with biochemical experiments performed from 2017 to 2018, and data were subsequently analyzed from 2019 to 2021. All participants, including healthy women and women with IBD, were enrolled at the University of Alberta affiliated pregnancy in IBD clinic (PregIBD, Edmonton, AB, Canada) between 2014 and 2017. Participants were recruited via referrals from primary care providers, providing gastroenterologists, and obstetricians. Inclusion criteria for patients with IBD required a diagnosis of UC or CD and age > 18 years. Patients were excluded if they had a history of indeterminate colitis, a history of surgical interventions including ileostomy, pouch, or previous bowel resections, or other autoimmune conditions such as SLE, rheumatoid arthritis, and multiple sclerosis. For healthy controls, inclusion criteria required age > 18 years and absence of IBD diagnoses. Control subjects were also excluded if they had other comorbidities, autoimmune, or inflammatory conditions, including and not limited to SLE, rheumatoid arthritis, and multiple sclerosis.

### 4.2. Cohort Follow-Up

The participants were followed using the schematic shown in Figure 1. Briefly, all subjects entered the study either at trimester 1 (T1, 0 to 13 weeks and 6 days) or trimester 2 (T2, 14 to 26 weeks and 6 days). Follow-up was done at the ambulatory PregIBD clinic at the following timepoints: T1, T2, trimester 3 (T3, 27 weeks to delivery), post-partum month 3 (PP3), post-partum month 6 (PP6), and post-partum month 12 (PP12). Patient outcomes collected include a combination of biological sampling and questionnaires for maternal disease outcomes, pregnancy outcomes, and neonatal outcomes.

### 4.3. Baseline Visits

Initial visits were conducted at the time of recruitment at T1 or T2. Patient demographics and clinical IBD characteristics were collected via questionnaires containing demographic, IBD, and medical variables. Demographic data included age, age of diagnosis, ethnicity, highest education obtained, employment, and smoking or alcohol use. IBD characteristics included IBD subtype, localization of disease, previous medications used, and current medications.

### 4.4. Follow-Up Visits

During subsequent visits, all patients were followed at the PregIBD clinic, and IBD disease status was tracked using pregnancy tracking questionnaires. IBD disease characteristics included current IBD and non-IBD medications, admissions to hospital, and clinical disease activity (measured using pMayo (partial Mayo) and mHBI (modified Harvey Bradshaw Index) clinical indices) evaluated by a trained Gastroenterologist. Pregnancy-related outcomes include complications including venous thromboembolism (VTEs), gestational diabetes, pre-eclampsia, and placental abnormalities such as placental abruption and placental previa, as well as admissions to the hospital. Maternal blood samples were collected each visit for serum cytokine panel analysis. Maternal stool samples were collected and flash-frozen for FCP analysis. Active disease was defined as FCP ≥ 250 µg/g or CRP ≥ 10 mg/L. Perinatal outcomes were collected at post-partum visits within 2-weeks of delivery via questionnaires or clinical records. Obstetrical variables included mode of delivery (vaginal versus C-section), assisted versus non-assisted vaginal delivery, elective versus emergency C-section, obstetrical complications (pre-eclampsia, placental abnormalities, or PPROM), offspring sex, birthweight, birth length, and admissions to NICU. Infants were not required to be brought to the PregIBD clinic for study visits.

### 4.5. Maternal FCP

Maternal stool samples were collected into sterile 25-mL vials. All samples were immediately collected by coordinators, barcoded, and stored at −80 °C until further analysis. For processing, stool samples were thawed, and FCP extracted using Buhlmann Calex Cap as per manufacturer protocols [45]. Briefly, the samples were vortexed and allowed to sit at room temperature for at least 10 min Samples were then loaded in duplicates onto 96-well plates using the FCP ELISA kits and quantified as per previous protocols [46].

### 4.6. Maternal Cytokine Panel

Human serum samples were assayed using V-PLEX pro-inflammatory Panel 1 human kit (Cat# K15049D), cytokines panel 1 human kit (Cat# K15050D), and Th17 panel human kit (Cat# K15085D, all derived from Meso Scale Discovery (Rockville, MD, USA) based on manufacturer’s protocol as previously described [47]. Briefly, maternal serum samples were stored at indicated timepoints during routine pregnancy bloodwork at pregnancy and post-partum timepoints. Serum samples were immediately barcoded and frozen at −80 °C. At the time of analysis, samples were thawed, and 25 µL was used using specified kits and quantified using the electro-chemiluminescent method in duplicates per sample. All chemiluminescent levels were quantified using a SECTOR S6000 plate reader (Meso Scale Discovery), and mean concentrations were derived from duplicate reads. Cytokine expression levels were expressed as absolute concentrations. Concentration outliers (defined as >3 standard deviations) were removed post-normalization. Visualization of the dataset in heatmaps was performed using TBtools [48].

### 4.7. Statistical Analysis

Statistical comparisons were conducted using GraphPad Prism version 6.0 (GraphPad Prism, GraphPad Softwares, San Diego, CA, USA) or Jamovi version 2.0. Cytokine data were expressed as mean ± standard errors. To detect differences between groups, an unpaired Student’s *t*-test was performed. For correlation modeling, Spearman and Pearson correlation coefficients were calculated. A *p*-value of < 0.05 was deemed statistically significant.

### 4.8. Study Approval

The study design was reviewed and approved by the institutional ethics board at the University of Alberta (protocol #Pr000056685). Informed written consent was received from all participants prior to participation in the study. All participants were followed from 2014 to 2017.

### 4.9. Data Availability

The datasets generated or analyzed in the current study are not made publicly available due to concerns about patient privacy but are available from the corresponding author upon reasonable request.

## 5. Conclusions

Nevertheless, to the best of our knowledge, this is the largest study to date to monitor the serum cytokines of a cohort of pregnant women with IBD. Our study demonstrates that maternal cytokines differ by disease phenotypes such as UC versus CD, disease activity, and medication used. Importantly, we show that the women with T1 elevation in serum IL-22 and IL-6 were more likely to develop flare-ups during pregnancy. Together, these results help clarify the immunological landscape of women with IBD in pregnancy and help identify IL-22 and IL-6 as independent predictors of flare-up during pregnancy. Further studies to substantiate and validate these findings are warranted to optimize the rational use of biologics to improve pregnancy outcomes.

## Figures and Tables

**Figure 1 ijms-23-10281-f001:**
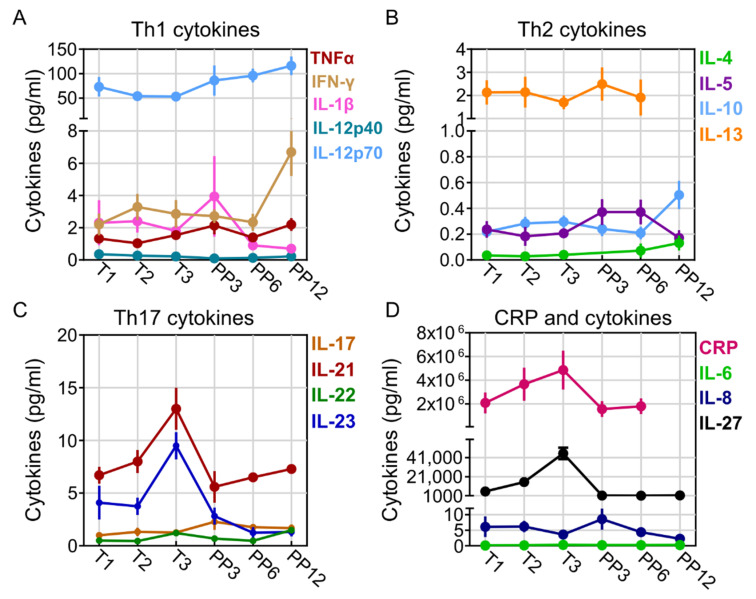
Pregnancy-related cytokine changes in healthy women. (**A**) Th1-cytokines TNFα, IFN-γ, IL-1β, IL-12p40, and IL-12p70 throughout pregnancy (N = 5–11 per time point). (**B**) Th2-cytokines IL-4, IL-5, IL-10, and IL-13 throughout pregnancy (N = 5–11 per time point). (**C**) Th17-cytokines IL-17, IL-21, IL-22, and IL-23 throughout pregnancy (N = 5–11 per time point). (**D**) CRP and cytokines IL-6, IL-8, and IL-27 throughout pregnancy (N = 5–11 per time point).

**Figure 2 ijms-23-10281-f002:**
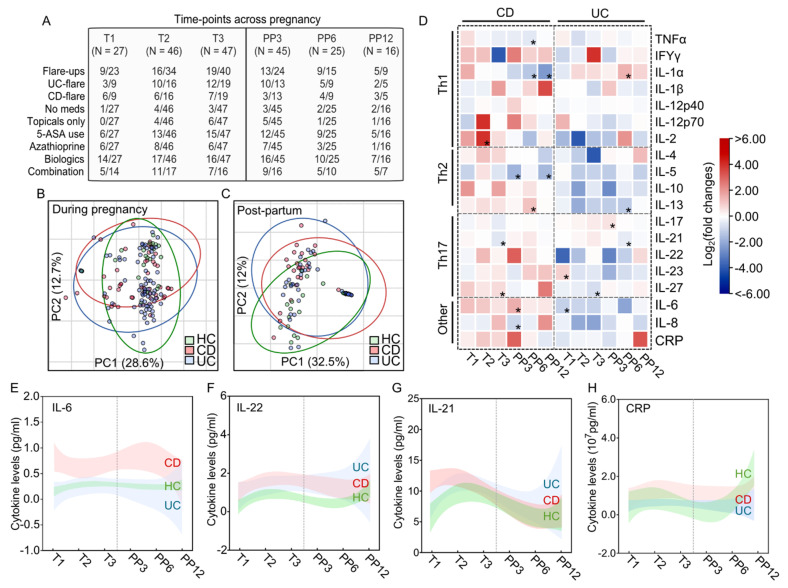
IBD status and subtype influence maternal serum cytokines throughout pregnancy. (**A**) IBD characteristics of the entire cohort of women with IBD by disease flare-ups and medications used (N = 16–47 per group). Principal coordinate analyses (PCA) of cytokine changes at (**B**) T2 and (**C**) PP3 by IBD disease subtype (N = 7–32 per group). (**D**) Heatmap demonstrating the log_2_ (fold changes) of serum cytokines across pregnancy (N = 16–47 per group). Logarithmic lines of best fit illustrating the cytokine variations across pregnancy for (**E**) IL-6, (**F**) IL-22, (**G**) IL-21, and (**H**) CRP. Shaded areas denote 95% confidence intervals. HC denotes healthy control. T1–3 denotes trimesters 1, 2, and 3, whereas PP3–12 signifies post-partum months 3, 6, and 12. * signifies *p* < 0.05 using unpaired Student’s *t*-test.

**Figure 3 ijms-23-10281-f003:**
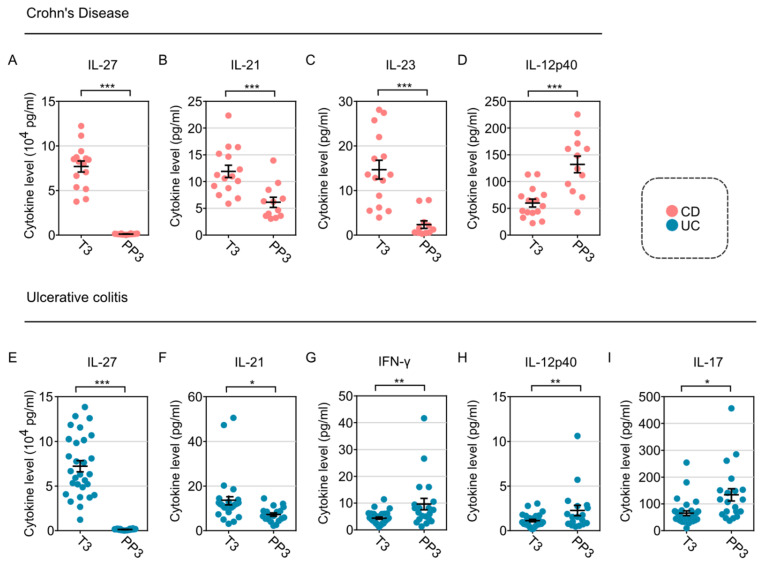
Cytokine changes altered by delivery in women with IBD. (**A**) IL-27, (**B**) IL-21, (**C**) IL-23, and (**D**) IL−12p40 were significantly altered between trimester 3 and post-partum month 3 in women with CD (N = 16–21 patients per cytokine tested). (**E**) IL-27, (**F**) IL-21, (**G**) IFN-γ, (**H**) IL-12p40, and (**I**) IL-17 were significantly altered between trimester 3 and post-partum month 3 in women with UC (N = 20–25 patients per cytokine tested). T3 denotes trimester 3 and PP3 signifies post-partum month 3. * signifies *p* < 0.05, ** signifies *p* < 0.01, and *** denotes *p* < 0.001 using unpaired Student’s *t*-test.

**Figure 4 ijms-23-10281-f004:**
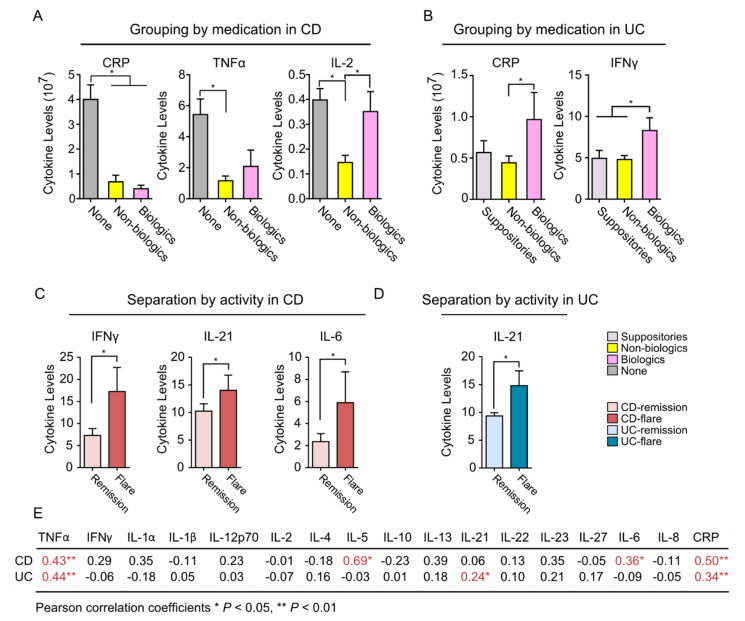
Maternal serum cytokines by medication use and active flare-up during pregnancy. (**A**) Maternal CRP, TNFα, and IL-2 differ by medication use in women with CD during pregnancy (N = 8–14 patients per group). (**B**) CRP and IFN-γ differ by medication use in women with UC (N = 7–22 patients per group). (**C**) IFN-γ, IL-21, and IL-6 are increased with active disease in women with CD (N = 29), whereas (**D**) only IL-21 was increased with active disease in women with UC during pregnancy. (**E**) Table illustrating the Pearson correlations between active disease (measured using either FCP or CRP cut-offs) and serum cytokines in women with IBD. Numbers represent Pearson correlation coefficients, * denotes *p* < 0.05, and ** denotes *p* < 0.01.

**Figure 5 ijms-23-10281-f005:**
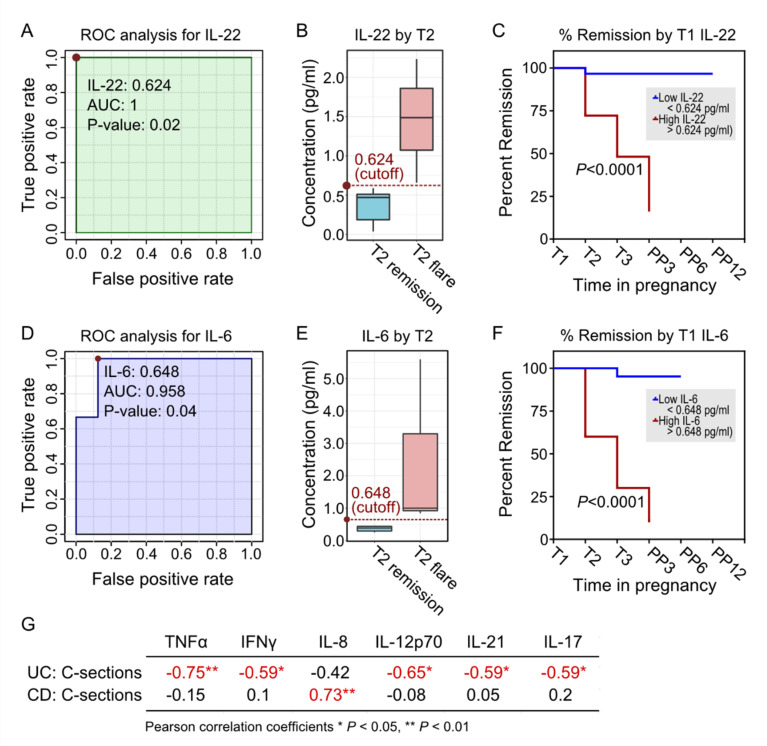
Predictive utility of T1 maternal serum cytokines in predicting the course of IBD disease during pregnancy. (**A**) ROC analysis using serum IL-22 levels at T1 at a cut-off of (**B**) 0.624 is associated with active disease activity at T2 in women with UC (AUC = 1, *p* = 0.02). (**C**) Percent remission curves of UC patients with known T1 IL-22 levels throughout pregnancy stratified based on IL-22 levels (N = 12). (**D**) ROC analysis using serum IL-6 levels at T1 at a cut-off of (**E**) 0.648 is associated with active disease activity at T2 in women with CD (AUC = 0.958, *p* = 0.04). (**F**) Percent remission curves of CD patients with known T1 IL-6 levels throughout pregnancy stratified based on IL-6 levels (N = 15). (**G**) Pearson correlation coefficients correlating T1 serum cytokines with obstetrical outcomes at time of delivery. * denotes *p* < 0.05, ** denotes *p* < 0.01.

**Table 1 ijms-23-10281-t001:** Baseline demographics of the study cohort.

	Healthy (N = 11)	CD (N = 29)	UC (N = 47)	*p*-Value
Age at conception (± SD)	31 ± 1.6	31 ± 4.7	30 ± 5.0	0.55 ^‡^
Age of diagnosis (± SD)	-	23 ± 6.1	23 ± 6.7	0.97 ^‡^
Marital Status				0.09 ^†^
Common-law	2/11 (18%)	10/27 (37%)	6/43 (14%)	
Married	9/11 (82%)	15/27 (56%)	28/43 (65%)	
Single	0/11 (0%)	0/27 (0%)	3/43 (7%)	
Ethnicity				0.52 ^†^
Caucasian	5/11 (45%)	22/27 (82%)	33/43 (77%)	
South Asian	1/11 (9%)	0/27 (0%)	2/43 (5%)	
Employment				0.65 ^†^
Full-time	8/11 (73%)	16/27 (59%)	27/43 (63%)	
Part-time	3/11 (27%)	3/27 (11%)	2/43 (5%)	
Education				0.78 ^†^
High school	0/11 (0%)	6/27 (22%)	7/43 (16%)	
Trades	0/11 (0%)	8/27 (30%)	11/43 (26%)	
University	6/11 (55%)	11/27 (41%)	19/43 (44%)	
Smoking status				0.88 ^†^
Ex-smoker	2/11 (18%)	12/27 (44%)	19/43 (44%)	
Alcohol use				0.68 ^†^
Ex-drinker	5/11 (45%)	17/27 (63%)	28/43 (65%)	
Never	6/11 (55%)	7/27 (26%)	9/43 (21%)	
Clinical flare-up	-	5/28 (18%)	24/40 (60%)	0.02 ^†^
Biochemical flare-up	-	12/20 (60%)	19/36 (53%)	0.84 ^†^
Prior pregnancies	4/10 (40%)	15/27 (55%)	29/46 (63%)	0.55 ^†^
Prior miscarriages	0/4 (0%)	8/15 (53%)	14/29 (48%)	0.16 ^†^
Prior elective abortions	1/4 (25%)	1/15 (7%)	1/29 (3%)	0.24 ^†^
Prior preterm delivery	0/4 (0%)	0/15 (0%)	0/29 (0%)	-
Prior pre-eclampsia	0/4 (0%)	2/15 (13%)	0/29 (0%)	0.10 ^†^
Prior placenta previa	0/4 (0%)	0/15 (0%)	1/29 (3%)	0.72 ^†^
Prior PROM	0/4 (0%)	0/15 (0%)	0/29 (0%)	-

^†^ Pearson. ^‡^ Wilcoxon.

**Table 2 ijms-23-10281-t002:** Pregnancy outcomes by IBD subtype.

Pregnancy Outcomes	CD (N = 29)	UC (N = 47)	*p*-Value
Preterm birth	3/29 (10%)	7/47 (15%)	0.57 ^†^
Vaginal deliveries	10/29 (34%)	26/43 (60%)	0.10 ^†^
C-section deliveries	19/29 (66%)	17/43 (39%)	0.10 ^†^
Induced labor	2/24 (8%)	11/43 (25%)	0.06 ^†^
Failed vaginal deliveries	2/24 (8%)	4/43 (9%)	0.80 ^†^
Emergency C-sections	8/24 (24%)	10/47 (23%)	0.82 ^†^
Planned C-sections	11/29 (38%)	7/47 (15%)	0.10 ^†^
Assisted deliveries	0/29 (0%)	2/47 (4%)	0.96 ^†^
Placental hemorrhage	0/29 (0%)	1/47 (2%)	0.72 ^†^
Congenital anomalies	0/29 (0%)	1/47 (2%)	0.72 ^†^
Neonatal infections	1/29 (3%)	0/47 (0%)	0.43 ^†^
Spontaneous abortions	1/29 (3%)	1/47 (2%)	0.96 ^†^
Pre-eclampsia	3/29 (10%)	0/47 (0%)	0.02 ^†^
Placental previa	0/29 (0%)	0/47 (0%)	-
PROM	2/28 (7%)	6/47 (13%)	0.56 ^†^
PPROM	2/28 (7%)	6/47 (13%)	0.56 ^†^
Placental abruptions	1/28 (3%)	0/47 (0%)	0.43 ^†^
Chorioamnionitis	1/25 (4%)	1/47 (2%)	0.43 ^†^
Gestational diabetes	1/29 (3%)	1/47 (2%)	0.93 ^†^
Venous thromboembolism	0/29 (0%)	0/47 (0%)	-

^†^ Pearson correlation coefficients.

**Table 3 ijms-23-10281-t003:** IBD medications and doses used in pregnancy.

Medications	CD (N = 29)	UC (N = 47)
5-ASA use Salofalk^®^ (2–4 g daily) Pentasa^®^ (4 g daily) Mezavant^®^ (1.2–2.4 g daily) Asacol^®^ (1.6–4.8 g daily)	1/26 (4%) 0/26 (0%) 1/26 (4%) 0/26 (0%)	15/47 (32%) 5/47 (11%) 6/47 (13%) 7/47 (15%)
Immunosuppressants Corticosteroids Imuran^®^ (150 mg daily)	1/26 (4%) 4/26 (16%)	0/47 (0%) 3/47 (6%)
Biologic use Infliximab Adalimumab Vedolizumab	5/26 (26%) 4/26 (0%) 0/26 (0%)	4/47 (9%) 2/47 (4%) 1/47 (2%)

## Data Availability

The datasets generated or analyzed in the current study are not made publicly available due to concerns about patient privacy but are available from the corresponding author upon reasonable request.

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
