# Peer review of "Elevated IL-6 and IL-22 in Early Pregnancy Are Associated with Worse Disease Course in Women with Inflammatory Bowel Disease"

_ijms, 2022, doi:10.3390/ijms231810281_

Round 1
Reviewer 1 Report
This is a prospective study evaluating the cytokine of pregnant IBD patients v.s. pregnant healthy subjects control. Active UC was found associated with elevation in IL-21. active CD was associated with elevated IFN-gamma, IL-6 and IL-21. IL-22 and IL-6 were assocatied with worsed IBD disease activity throughout pregnancy in IBD patients.The findings are useful for further clinical management of pregnant IBD patients. Some issues may need further clarification.
1. The demographic data were matches well between groups. Is there any information about previous pregnancy and delivery histories? Or all patients were having first time of pregenancy in this study?
2. Low vaginal deliveries were observed in CD group. As patients having surgery history were excluded in this study, what is the poteintial cause?
3. In table 2 , the p value of emeregency C- sections were 0.02 with only 3% difference. Please reconfirm about it.
4. The cytokine may also be influenced by the medication use. Is the phenomemon that increased cytokine which were found with sorsed IBD disease activity observed in all patients, or only patients not taking biologics? The current subroup analysis seemed to show that the changes of IFN-gamma, IL-6, IL-21, IL-22 all dissapear after adjustment of IBD MEDICAITON?
5. As the authors found observed that CRP was consistently elevated across pregnancy, is it related to disease activity or nature presensation of pregnancy?
6. According to the study's finding, physicians may check pregnant IBD patients' cytokine to predict the pregnenancy outcome in the future? If elevated IL-22, IL-6 found in future patients, what is the authours suggestion?
Author Response
Review #1
- The demographic data were matches well between groups. Is there any information about previous pregnancy and delivery histories? Or all patients were having first time of pregnancy in this study?
We agree that this information would be crucial to the interpretation of results, and have added additional data on prior pregnancy including prior history of any pregnancies, miscarriages, elective abortions, preterm delivery, pre-eclampsia, placental previa and PROM (see Table 1).
- Low vaginal deliveries were observed in CD group. As patients having surgery history were excluded in this study, what is the potential cause?
We have re-analyzed the pregnancy outcomes and statistics in Table 2. In the revised table, although we still observe a decreased proportion of CD patients undergoing vaginal deliveries, this did not reach statistical significance. This difference in C-section rates between UC and CD mothers has been described prior (PMID: 34046084).
- In table 2, the p value of emergency C-sections were 0.02 with only 3% difference. Please reconfirm.
Thank you for flagging this error. We have re-analyzed the entire datasets and statistics in Table 2 and noted that prior analysis did not cater to missing data for each variable measured. This has been revised accordingly (see Table 2).
- The cytokine may also be influenced by the medication use. Is the phenomenon that increased cytokine which were found with worse IBD disease activity observed in all patients, or only patients not taking biologics?
The subgroup analysis was done separately for medications and disease activity against all patients as there were not enough sample size for further stratification with both variables. We have revised the wording to reflect this difference (see Lines 222 – 224). The Figure 4 legend has also been revised to signify ** denotes P < 0.01.
- As the authors found observed that CRP was consistently elevated across pregnancy, is it related to disease activity or nature presentation of pregnancy?
This is related to physiological changes in pregnancy as shown in Figure 1D which was done only using datasets from the healthy controls, show a persistent rise in CRP throughout pregnancy period. The wording in the results section has been clarified to make this point (see Line 182 - 183).
- According to the study's finding, physicians may check pregnant IBD patients' cytokine to predict the pregnancy outcome in the future? If elevated IL-22, IL-6 found in future patients, what is the author’s suggestion?
While we are excited about these results, our study is preliminary and subjected to small sample size and narrow spectrum of disease. We suggest that larger studies will be needed to replicate our findings, this is now clarified in discussion (see Lines 336 – 340).
Reviewer 2 Report
Wu et al provide a valuable data set that describes a longitudinal cytokine alteration during the pregnancy in IBD patients. The data included in this paper could be an asset in providing a comprehensive picture for future study design, therefore sharing the data and the meta-data as supplementary tables is strongly recommended. Some of the points addressed by the authors are fairly clear, but some connections and conclusions are not fully substantiated. There are problems with discussions and data analyses. Particularly, regarding the treatment and the pregnancy sage, it is really unclear whether the medication condition was corrected in the analyses comparing T3 and PP3, as well as the whether the fluctuation of cytokine levels during the pregnancy was controlled/corrected in the comparisons between different medications. It is not a fair comparison comparing cytokine levels from patients with treatment A at T1 to treatment B at T3.
Some other minor suggestions:
1) Table 1: include SD in the age rows
2) Fig 1: (A)-(D): How do the plots look if raw concentration instead of the normalized values are used in the plot? (E)-(G) It’s not clear how the normalization of the expression level and how the combination of different cytokines for Th1/Th2/Th17 were conducted? It’s really unclear how to get (E)-(G) from (A)-(C)
3) Fig 2: (B)-(C) difficult to see the color of the dots. Plot (D) using -log2FC instead of log2FC is really confusing for the readers particularly together with plot (E)-(H) as the increased levels in UC or CD versus HC will be negative in (D) but shown as higher lines in (E)-(H). If there is no particular reason to use -log2FC in (D), I would recommend to use log2FC.
4) Fig 3: It would be great to also include HCs as a supplementary figure describing the changes for these cytokines (or refer to figure 1 if the measured values are used)
5) Fig 4: Chord plots did not provide more but less information than a table. Visualization by the Chord plot in this case also makes the interpretation more difficult. Readers will only get inaccurate feelings of the relative strength of the correlation (in this case all correlation coefficients were added up to the full circumference) but not the direction (the sign) and the exact value of the correlation coefficients.
6) Figure 5: (G) the color gradient and the difference in size are unclear.
Author Response
Reviewer #2
- Particularly, regarding the treatment and the pregnancy sage, it is really unclear whether the medication condition was corrected in the analyses comparing T3 and PP3, as well as the whether the fluctuation of cytokine levels during the pregnancy was controlled/corrected in the comparisons between different medications. It is not a fair comparison comparing cytokine levels from patients with treatment A at T1 to treatment B at T3.
We agree with the suggestion that the comparison is better if medications and fluctuations in cytokines have been controlled in the comparison. This was not possible given the small sample size we have available during subgroup analysis. We feel this data will still be useful but these limitations have been noted in the Discussion now (Lines 360 – 365).
- Table 1: include SD in the age rows
The standard deviations have been added as suggested (see Table 1).
- Fig 1: (A)-(D): How do the plots look if raw concentration instead of the normalized values are used in the plot? (E)-(G) It’s not clear how the normalization of the expression level and how the combination of different cytokines for Th1/Th2/Th17 were conducted? It’s really unclear how to get (E)-(G) from (A)-(C)
We agree with this point and have remade the entire Figure 1 with the raw concentrations instead, please see Figure 1. Please note that given our cytokine panel only contains 20 or so targets, we cannot fully describe a Th1/Th2/Th17 response in panels E)-G). We have therefore removed analyses E)-G) as well.
- Fig 2: (B)-(C) difficult to see the color of the dots. Plot (D) using -log2FC instead of log2FC is really confusing for the readers particularly together with plot (E)-(H) as the increased levels in UC or CD versus HC will be negative in (D) but shown as higher lines in (E)-(H). If there is no particular reason to use -log2FC in (D), I would recommend to use log2FC.
Thank you for the suggestion, this has been rechecked, and the panel is now represented as Log2(FC) as suggested. Dot plot figures have been completely removed and switched to a heatmap because of the later comments below about dot plots being difficult to visualize on our figure given small variations.
- Fig 3: It would be great to also include HCs as a supplementary figure describing the changes for these cytokines (or refer to figure 1 if the measured values are used).
We have re-added the absolute concentrations for the cytokines in healthy control as opposed to using normalize concentrations.
- Fig 4: Chord plots did not provide more but less information than a table. Visualization by the Chord plot in this case also makes the interpretation more difficult. Readers will only get inaccurate feelings of the relative strength of the correlation (in this case all correlation coefficients were added up to the full circumference) but not the direction (the sign) and the exact value of the correlation coefficients.
We agree with this suggestion and have replaced the chord diagram with a table for the correlation coefficients, please see revised Figure 4.
- Figure 5: (G) the color gradient and the difference in size are unclear.
We agree with this suggestion and have replaced the dot plot with a table for the correlation coefficients, please see revised Figure 5.
Reviewer 3 Report
Wu et al. performed a longitudinal profiling of multiple cytokines in pregnant women with IBD. This is a large cohort study and is difficult one to conduct. The authors identified various cytokine patterns associated with disease activity and stages of pregnancy. This an interesting study and a valuable one to the IBD literature.
The following are the comments to improve the manuscript.
Major comments:
In table 1, please check the numbers in smoking status as they appear redundant. The total numbers are not adding up for most of the parameters. Were these due to missing data?
Why were the pregnancy outcomes not compared with healthy subjects? Was this due to low numbers in that group or any other reason?
The statistics was given as Students t-test for between groups. Was that paired t-test analysis?
There were 76 (29+47) IBD patients in the study, but the maximum number of subjects for serum cytokines was 60 at T2 (L191). What was the reason for this? If all the data points were not derived from each subject, then how did the authors normalize or compare the data?
The authors claim this is one of the largest cohorts of 74 women with IBD (L357). It is unclear why the numbers are not consistent.
The authors classified IL-2 as Th2 cytokine. IL-2 is not one of the typical Th2 cytokines unless the authors have strong evidence to do so.
There is no IL-27 data for the healthy individuals. Was IL-27 not detectable or not tested in these individuals?
In figure 2E, the cytokine levels were given as ug/ml for all of them. Since IL-6 levels are usually expressed in pg/ml, please check all the units including in other figures and data presented. This is very important as future studies may compare these values and hence verify all values and units.
In figure 4 legend, the number of UC patients was given as 51 while the study has only 47 of them. Please check all the numbers wherever applicable to avoid any errors.
Minor comments:
L48: IL-12 was entered in Th2 response category. Was this supposed to be IL-13 instead of IL-12? Also note the reference cited was from 2002, hence change the citation to a newer one as there is a vast knowledge acquired in the last 20 years.
Please expand mHBI and FCP. Fecal calprotectin is given in the abstract and it would be helpful if this expansion provided at the first instance of FCP.
Please provide the catalog number for kits used in this study which will help if anyone wants to reproduce or conduct similar research.
Marital status is entered as martial status in table 1. More language corrections are required in the manuscript.
Author Response
Reviewer #3:
- In table 1, please check the numbers in smoking status as they appear redundant. The total numbers are not adding up for most of the parameters. Were these due to missing data?
This had been checked and corrected. These were due to missing data as not all variables had 100% of respondence rates. The denominators were of patients who responded.
- Why were the pregnancy outcomes not compared with healthy subjects? Was this due to low numbers in that group or any other reason?
The healthy subjects had low numbers to truly compare event rates between groups.
- The statistics was given as Students t-test for between groups. Was that paired t-test analysis?
The statistics done was Unpaired t-test. This information has now been added into the methods and corresponding legends in all figure legends.
- There were 76 (29+47) IBD patients in the study, but the maximum number of subjects for serum cytokines was 60 at T2 (L191). What was the reason for this? If all the data points were not derived from each subject, then how did the authors normalize or compare the data?
This was due to not all patients being able to provide cytokine samples at all timepoints. This was particularly significant from post-partum months 6 and onwards. We have made this point clearer in our Discussion to highlight the incomplete datasets from all patients (see Lines 254 – 261).
- The authors claim this is one of the largest cohorts of 74 women with IBD (L357). It is unclear why the numbers are not consistent.
This was a typo and has been corrected as pointed out (see Line 357).
- The authors classified IL-2 as Th2 cytokine. IL-2 is not one of the typical Th2 cytokines unless the authors have strong evidence to do so.
The role of IL-2 is more thought to be as a Th1 cytokine, we have made this change in corresponding figures, please see revised Figure 3D.
- There is no IL-27 data for the healthy individuals. Was IL-27 not detectable or not tested in these individuals?
IL-27 has now been added into revised Figure 1, which now also contains the absolute concentrations of cytokines in the healthy controls as opposed to normalized values as suggested by another reviewer. Corresponding results sections are updated (Lines 180 – 183).
- In figure 2E, the cytokine levels were given as ug/ml for all of them. Since IL-6 levels are usually expressed in pg/ml, please check all the units including in other figures and data presented. This is very important as future studies may compare these values and hence verify all values and units.
Thank you for the suggestion, we have rechecked all the units and the units have been corrected to pg/ml. This has been verified and corrected in all figures.
- In figure 4 legend, the number of UC patients was given as 51 while the study has only 47 of them. Please check all the numbers wherever applicable to avoid any errors.
This was a typo and has been corrected to 47, please see revised Figure 4 Legend.
Minor comments:
- L48: IL-12 was entered in Th2 response category. Was this supposed to be IL-13 instead of IL-12?
This was corrected.
- Please expand mHBI and FCP. Fecal calprotectin is given in the abstract and it would be helpful if this expansion provided at the first instance of FCP.
As suggested, we explained the abbreviations (see Line 21, and mHBI and pMayo now elaborated in results section).
- Please provide the catalog number for kits used in this study which will help if anyone wants to reproduce or conduct similar research.
We have provided this now in the methods section (see Lines 130 – 133).
- Marital status is entered as martial status in table 1. More language corrections are required in the manuscript.
This correction has been made, see revised Table 1.
Reviewer 4 Report
This pilot study is measuring maternal serum cytokines longitudinally from the first trimester of the gestational week to 12 months postpartum in a cohort of 76 pregnant women with IBD. To explore factors influencing host cytokine changes, cytokine changes were stratified based on IBD status, IBD subtype, disease flare-ups, and drugs used at the time of sampling. In addition, to explore the clinical utility of cytokine levels, serum cytokines and disease activity were correlated across time points to test the predictive utility of early pregnancy serum cytokine changes for future flare-ups and pregnancy-related outcomes during pregnancy.
Comments are indicated below.
Overall
In this study, did you investigate the association between each cytokine level and pregnancy outcomes?
Abstract
'Inflammatory bowel diseases' is abbreviated as IBD. This means that the beginning of the phrase must be capitalized. All abbreviations should also be spelled out on the first appearance.
I think the wording is inappropriate.
“trimester one to 12 months post-partum”: From the first trimester of pregnancy to the first month postpartum.
Why was the erythrocyte sedimentation rate not assessed?
Results
Line 180: “CRP” had already appeared in the Methods section.
Diurnal variations exist in blood cytokine levels, but were all samples measured at the same time?
Considers that drug therapy during pregnancy should be included in the patient background table, including drug name, dose, route of administration, etc.
Author Response
Reviewer #4
- In this study, did you investigate the association between each cytokine level and pregnancy outcomes?
Thank you for the suggestion, we have added the results for all cytokines and included the Pearson correlation coefficients in Table form (see Supplemental Tables 3 and 4). We also changed this panel to a table format as per Reviewer #5’s suggestions on clarity and rephrased the results section to clarify this point (see Lines 256 – 263).
- Abstract: 'Inflammatory bowel diseases' is abbreviated as IBD. This means that the beginning of the phrase must be capitalized. All abbreviations should also be spelled out on the first appearance.
This has been corrected (see Line 15).
- I think the wording is inappropriate. “trimester one to 12 months post-partum”: From the first trimester of pregnancy to the first month postpartum.
This has been corrected (see Lines 19 – 20).
- Why was the erythrocyte sedimentation rate not assessed?
While useful as an acute phase reactant, ESR was not used as it is not a consistent marker for disease activity as CRP and FCP, and suffers from variations due to anemia, and its variation does not often correlate with inflammation (Vermeire et al, Gut 2006, PMID: 1647109).
- Line 180: “CRP” had already appeared in the Methods section.
This has been now corrected (see Line 182).
- Diurnal variations exist in blood cytokine levels, but were all samples measured at the same time?
The samples were only measured at the same time but collection times were not controlled. We recognize this can cause variations and have added this point to Discussion (see Lines 368 – 370).
- Considers that drug therapy during pregnancy should be included in the patient background table, including drug name, dose, route of administration, etc.
Thank you for the suggestion, we have added Table 3 to show the brand names and doses as suggested. However, for the biologics not all dosing were recorded and in recognition of this as a limitation, we have added the issue of biologic does and drugs level to the Discussion (see Lines 363 – 370).
Reviewer 5 Report
the authors were interested in the cytokine profiles of IBD and CD patients, in remission or not at conception. They provide new and interesting data, for those diseases but also for healthy women. This may help in the follow-up of IBD abd CD patients, but also ti gain insight the understanding of IBD and CD.
Author Response
Review #5
- The authors were interested in the cytokine profiles of IBD and CD patients, in remission or not at conception. They provide new and interesting data, for those diseases but also for healthy women. This may help in the follow-up of IBD and CD patients, but also to gain insight the understanding of IBD and CD.
We thank the reviewer for the helpful comment.
Round 2
Reviewer 1 Report
The authors answered all my questions. I have no further questions.
Author Response
Thank you to the peer-reviewer for the helpful comments.
Reviewer 2 Report
1. figure 2: the x-axis (column title) of the heatmap is missing
2. As replied by the authors "We agree with the suggestion that the comparison is better if medications and fluctuations in cytokines have been controlled in the comparison. This was not possible given the small sample size we have available during subgroup analysis". Thus, it is crucial for interpreting the paper that readers can access data when needed because of this limitation. A supplementary table with all individual cytokine measurements at all time points should be provided or available upon request (please do mention this in the data availability statement).
Author Response
1. figure 2: the x-axis (column title) of the heatmap is missing
Thank you for the comment. We have revised the figure 2 and added the label for the heatmap.
2. As replied by the authors "We agree with the suggestion that the comparison is better if medications and fluctuations in cytokines have been controlled in the comparison. This was not possible given the small sample size we have available during subgroup analysis". Thus, it is crucial for interpreting the paper that readers can access data when needed because of this limitation. A supplementary table with all individual cytokine measurements at all time points should be provided or available upon request (please do mention this in the data availability statement).
Thank you, as suggested, we have included the option to provide the datasets upon request to the corresponding author. We have added a data availability statement and section for this. Please see Lines 155 - 158.